# Exploring in Extremely Dark: Low-Light Video Enhancement with Real Events

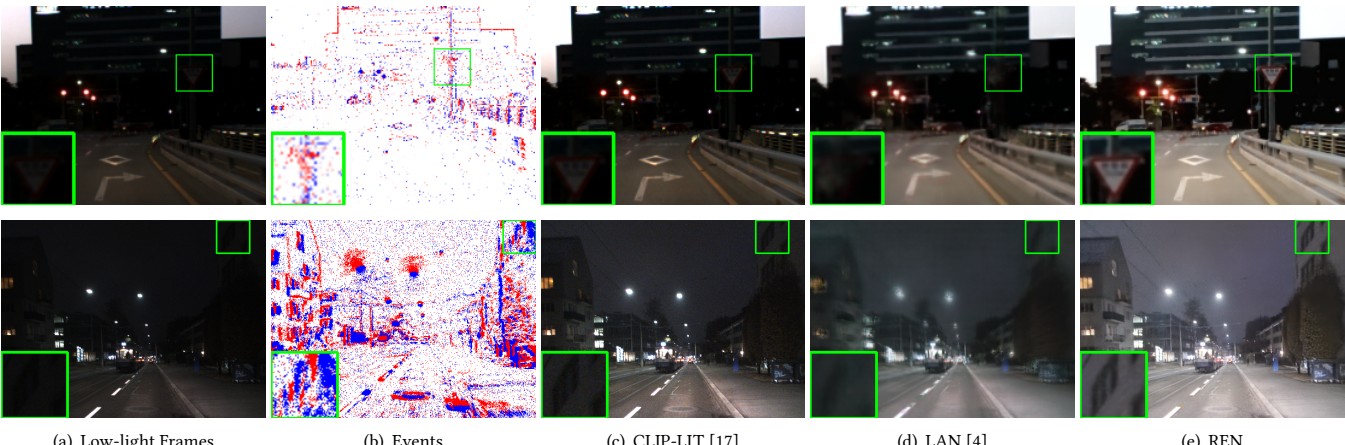

(a) Low-light Frames     (b) Events     (c) CLIP-LIT [17]     (d) LAN [4]     (e) REN

Figure 1: Given inputs of (a) low-light frames and (b) events, the normal-light frames can be enhanced using different methods shown in (c) CLIP-LIT (ICCV2023) [17], (d) LAN (ICCV2023) [4] and (e) the Real-Event Embedded Network (REN). These examples are tested on real data containing extremely dark regions. The REN is able to generate normal-light frames with more details and improved brightness.

## ABSTRACT

Due to the limitations of sensor, traditional cameras struggle to capture details within extremely dark areas of videos. The absence of such details can significantly impact the effectiveness of low-light video enhancement. In contrast, event cameras offer a visual representation with higher dynamic range, facilitating the capture of motion information even in exceptionally dark conditions. Motivated by this advantage, we propose the Real-Event Embedded Network for low-light video enhancement. To better utilize events for enhancing extremely dark regions, we propose an Event-Image Fusion module, which can identify these dark regions and enhance them significantly. To ensure temporal stability of the video and restore details within extremely dark areas, we design unsupervised temporal consistency loss and detail contrast loss. Alongside the supervised loss, these loss functions collectively contribute to the semi-supervised training of the network on unpaired real data. Experimental results on synthetic and real data demonstrate the superiority of the proposed method compared to the state-of-the-art methods. Our codes will be publicly available.

## CCS CONCEPTS

• **Computing methodologies → Neural networks**; **Machine learning algorithms**; **Reconstruction**; **Image Processing**.

## KEYWORDS

Low-Light, Video Enhancement, Real Event, Extremely Dark

## 1 INTRODUCTION

Low-light video enhancement is a critical task in the field of computer vision, which aims to enhance the visibility and visual quality of videos captured under low-light conditions. Due to the constraints of low-light environments and the limitations of camera sensor, these videos often suffer from noticeable noise, insufficient information and blurriness, leading to the loss of details and a decrease in the overall quality. Despite significant progress in this field, low-light video enhancement remains a formidable challenge.

Many deep learning methods [1, 4, 11, 21, 34, 43] have been proposed to enhance low-light videos. However, most of these approaches rely on mining information directly from a video to enhance it. Due to the limitations of cameras, details in extremely dark areas of the videos are inevitably lost and remain missing even after the enhancement.

Event camera is an emerging sensor characterized by its large dynamic range and rapid response. Events recorded alongside low-light videos, even in extremely dark areas, can fully capture motion information, which is lacking in traditional cameras. This feature makes event cameras highly suitable for low-light video enhancement.

However, employing event cameras for low-light video enhancement faces several distinct challenges: 1) Fusion complexities: The information in the extremely dark regions of the video is limited, making it challenging to fuse event data with these areas and achieve effective enhancement. 2) Data acquisition difficulties: Constructing a dataset consisting of paired real events, normal-light videos and low-light videos is difficult. Most existing video-event paired datasets [6, 14] only consist one type of videos while also suffering from alignment issues. While event simulators [10, 27] can be used to generate synthetic events, the inherent disparities between synthetic and genuine data can compromise the inferential performance on real data.

In this paper, we propose the **R**eal-Event **E**mbedded **N**etwork (REN), a low-light video enhancement model that integrates event data based on the Retinex [13] theory. We tackle the challenges by the following strategies: To effectively fuse misaligned real events and images, we present Event-Image Fusion (EIF) module. It can adaptively identify extremely dark regions and utilize event information to enhance them. Furthermore, EIF module establishes spatial alignment between misaligned image and event features. To address the challenge of lacking genuine paired data, we propose unsupervised temporal consistency loss and detail contrast loss, which offer supervision at both temporal and spatial levels. They ensure the temporal consistency of input and output sequence, and spatially supervise detail reconstruction in extremely dark regions respectively. Together with the supervised loss, these loss functions collectively contribute to the semi-supervised training of the REN. The overall network is trained end-to-end with partially labeled datasets, which consist of paired low-light and normal-light videos with synthetic events, as well as solely low-light videos alongside real events.

In summary, the contributions of our work are as follows:

- We propose the Real-Event Embedded Network to leverage real events for addressing low-light enhancement challenges. To the best of our knowledge, it is the first exploration of real events in this field.
- We propose the Event-Image Fusion module, which can identify extremely dark regions and utilize event to enhance them.
- We design unsupervised temporal consistency loss and detail contrast loss, aiming to maintain temporal stability of videos and restore details in extremely dark regions.
- Extensive experiments on both real and synthetic datasets demonstrate that our approach outperforms state-of-the-art methods in low-light video and image enhancement.

## 2 RELATED WORK

**Low-Light Video Enhancement** This line of work primarily includes direct enhancement and Retinex-based enhancement. In the former, Liu et al. [18] employ neural networks to simulate event information from videos and fuse them. Zhang et al. [40] extract optical flow from individual images to enforce the temporal stability. Chhirolya et al. [2] propose a self-cross dilated attention module to learn inter-frame relationships in static videos. SGLLIE [43] extracts enhancement factors and recursively enhances videos. MBLLEN [21] and SMOID [11] employ 3D convolutions to harness

the temporal information within videos. As for the Retinex-based enhancement, LAN [4] and SDSDNet [34] extract illumination from videos and enhance them, with LAN designing an adaptive brightness enhancement method capable of achieving one-to-multiple enhancement. In contrast to previous Retinex-based approaches, our focus lies on the reflectance, using event data to complement missing information in extremely dark regions.

**Event-Based Video Reconstruction** Event cameras have found widespread applications in the field of image and video reconstruction, including super-resolution [8, 20], high dynamic range [25, 39], deblurring [30, 36] and video frame interpolation [12, 31]. Considering the distinct modalities of image and event features, these works primarily focus on the fusion. In addition to simply summing or concatenating events and images [26, 33], HDRev [39] propose a multimodal representation alignment strategy to learn a shared latent space and a fusion module tailored to complement two types of signals in different regions. EvIntSR [8] constructs latent frames separately from events and images, followed by their fusion. EFNet [29] propose an Event-Image Cross-modal Attention fusion module, which allows attending to the event features via a channel-level attention mechanism. Very recently, Liang et al. [16] propose a low-light video enhancement method with hybrid inputs of events and frames, using synthetic events to train their network. The differences between these methods and our work are: 1) We employ real event for low-light video enhancement; 2) Our work focuses on enhancing extremely dark regions using event data, while other approaches treat all regions uniformly; 3) Our approach can deal with the misalignment image and event features through spatial-level attention.

## 3 PROPOSED METHOD

### 3.1 Formulation

**Event Representation** An event data $e = (p, t, \sigma)$ is triggered at time $t$ and at pixel $p = (x, y)$ when the log intensity change $\Delta L$ between $t$ and $t - \delta t$ exceeds the dispatched threshold $\theta$, $\sigma \in \{+1, -1\}$ is the polarity that indicates the increase or decrease of intensity changes.

$$\Delta L = \| \log R_t^{(p)} - \log R_{t-\delta t}^{(p)} \| \tag{1}$$

where $R_t^{(p)}$ denotes the instantaneous intensity at time $t$ and pixel $p$.

As event streams are sparse points and can not input into network, we represent the events as 3D voxel grid [44]. By discretizing duration $\Delta t = t_{K-1} - t_0$ spanned by $K$ events into $C$ temporal bins, each event $e_k = (p_k, t_k, \sigma_k)$ distributes its polarity $\sigma_k$ to the two closest voxels:

$$E_t^{(p)} = \sum_{p_k = p} \sigma_k \max(0, 1 - | t - \widetilde{t_k} |) \tag{2}$$

where $\widetilde{t_k} = \frac{C-1}{\Delta t}(t_k - t_0)$ is the normalized timestamp.

**Problem Statement** Classic Retinex-based model assumes that image can be decomposed into reflectance and illumination, similar to [3], we employ a regularization-free method to represent the image $I$:

$$I = \phi_s(\phi_R(I), \phi_L(I)) \tag{3}$$

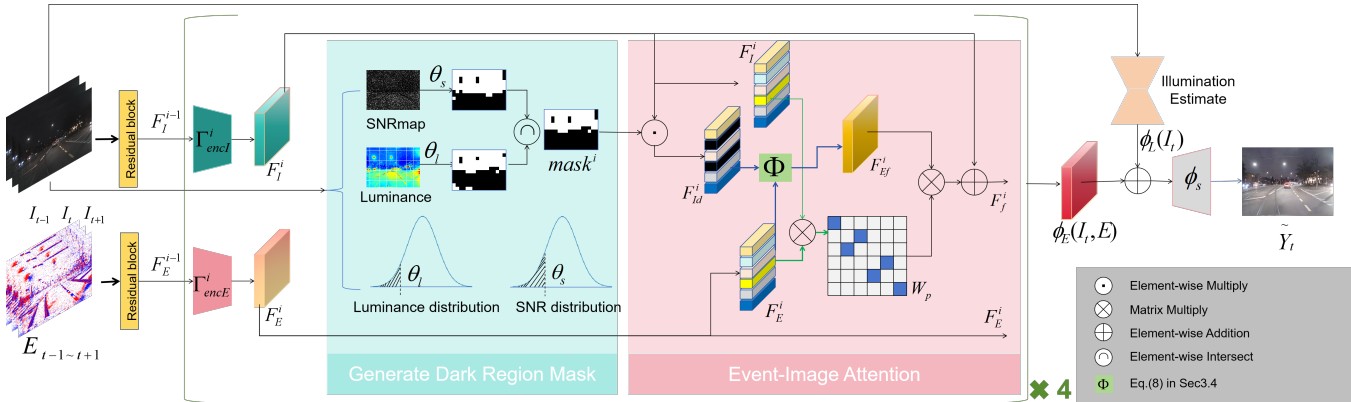

**Figure 2: An overview of the proposed REN. Based on the Retinex theory, estimate the illumination and reflection of input frames. Generate Dark Region Mask to identify the extremely dark regions, then fuse event and image features to enhance the reflection in these regions using the Event-Image Attention module. Finally, merge the illumination and reflection to obtain the enhanced frame.**

where $\phi_R$, $\phi_L$ denote the modules used to estimate reflectance and illumination respectively, and $\phi_s$ denotes the module used to synthesize the reflectance and illumination.

Previous methods primarily focus on accurately estimating these two components and their fusion, often directly assuming the estimated reflectance is same as the ground truth [3, 4]. However, in extremely dark regions, the reflectance tends to lack fine details, owing to the limitations of conventional camera sensors. Different from prior work, our approach enhances the reflectance using event data, the enhanced image $\widetilde{Y}_t$ can be defined as:

$$\widetilde{Y}_t = \phi_s(\phi_E(I_l, E), \phi_L(I_l)) \tag{4}$$

and the objective function is as follows:

$$\min_{\phi_s, \phi_L, \phi_E} \| I_n - \phi_s(\phi_E(I_l, E), \phi_L(I_l)) \|_F^2 + \\ \| \phi_R(I_n) - \phi_E(I_l, E) \|_F^2 \tag{5}$$

where $I_n$, $I_l$ and $E$ denote normal-light image, low-light image and event respectively; $\phi_E$ denotes the reflectance estimated from events and images; and $\| \cdot \|_F$ represents the Frobenius norm.

## 3.2 General Architecture of REN

Fig. 2 illustrates the overall architecture of the proposed REN. Given input video frames $I_{t+j}, j \in [-k,k]$ and an event sequence $E_{t-k\sim t+k}$, we first concatenate and project them as embedding $F_I^{(0)} \in R^{H \times W \times C}$ and $F_E^{(0)} \in R^{H \times W \times C}$ respectively through a Residual Block [9]. Then we estimate the reflectance $\phi_E(I_t, E)$ and illumination $\phi_L(I_t)$ from them, and finally synthesize them into normal light frames $\widetilde{Y}_t$.

We use a dual-branch structure to estimate the reflectance $\phi_E(I_t, E)$, with a branch dedicated to extract image features and the other to extract the event features respectively. Every branch is structured as a hierarchy with four stages. Each stage of the image branch consists of an image encoder $\Gamma_I^i$ and an Event-Image Fusion module (EIF, will be detailed in Sec. 3.3). Each stage of event branch only consists of an event encoder $\Gamma_E^i$. For the $i$-th stage, the image feature $F_I^{i-1} \in R^{\frac{H}{2^{i-1}} \times \frac{W}{2^{i-1}} \times 2^{i-1}C}$ and event feature $F_E^{i-1} \in$

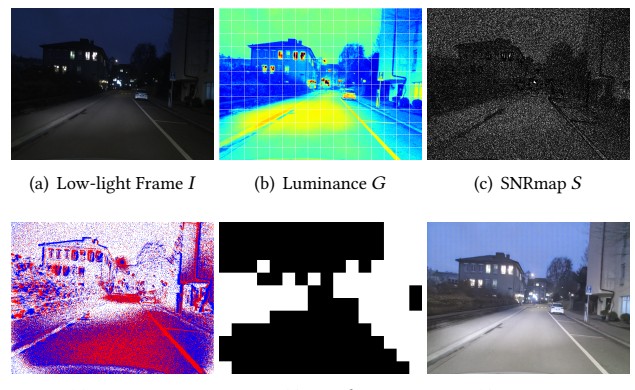

(a) Low-light Frame $I$    (b) Luminance $G$    (c) SNRmap $S$

(d) Event $E$    (e) *Mask*    (f) Our result

**Figure 3: Visualization of the Luminance (b) and SNRmap (c) extracted from input frame (a). The extremely dark regions mask is (e), the input events is (d) and our result is (f).**

$R^{\frac{H}{2^{i-1}} \times \frac{W}{2^{i-1}} \times 2^{i-1}C}$ will first be processed as $F_I^i \in R^{\frac{H}{2^i} \times \frac{W}{2^i} \times 2^i C}$ and $F_E^i \in R^{\frac{H}{2^i} \times \frac{W}{2^i} \times 2^i C}$ through image encoder $\Gamma_I^i$ and event encoder $\Gamma_E^i$ respectively. Then these features are jointly fed into the EIF module to obtain the fused features $F_f^i \in R^{\frac{H}{2^i} \times \frac{W}{2^i} \times 2^i C}$, we set $F_I^i = F_f^i$ as the input for the next stage. $F_f^4$ will be considered as the reflectance $\phi_E(I_t, E)$.

Considering that the illumination requires more global features, we first encode $F_I^{(0)}$ into the latent space, followed by a U-Net [28] to estimate the Illumination $\phi_L(I_t)$.

The estimated illumination $\phi_L(I_t)$ and reflectance $\phi_E(I_t, E)$ are first aligned by convolutional layers, after which the aligned features are fed into the decoder $\phi_s$ to synthesize them. In addition, we also employ residual connections between image encoders and the decoder to improve the performance. In the end, the output of the decoder is processed through a convolutional layer to obtain the enhanced frame $\widetilde{Y}_t$.

## 3.3 Event-Image Fusion

The fusion of image features and event features is the key to event-based low-light enhancement, requiring the matching of relevant features from both the image and the event. Due to the limited information in the ultra-dark regions and interference from other areas, matching is very challenging.

**Generate Dark Region Mask** We design a patch-based extremely dark region mask to identify these regions, so the constraints imposed by the two conditions mentioned above can be relaxed when matching with events. As the extremely dark regions are included in the patch, information within the patch can be used for matching with event features, reducing the difficulty of matching, meanwhile avoiding interference from information outside the patch.

Due to the extremely low brightness and Signal-to-Noise Ratio (SNR) in the ultra-dark regions, we use SNR distribution and luminance distribution to generate the mask. We calculate the SNR map using the method similar to [37], and regard the grayscale image as luminance map. Specifically, given the frames $I_{t+j, j \in [-k,k]}$, we calculate the luminance map $G$ and SNR map $S$ of $I_t$, and then count their distribution.

$$G = Gray(I_t)$$
$$\widetilde{I_t^g} = denoise(Gray(I_t))$$
$$S = \frac{\widetilde{I_t^g}}{abs(I_t^g - \widetilde{I_t^g})} \qquad (6)$$

where $Gray$ denotes the process of obtaining a grayscale image, $denoise$ represents a non-learning-based denoising operation. For the $i$-th stage, $F_I^i \in R^{h \times w \times c}$, we first resize $G$ and $S$ to $(h, w)$, then divide them into non-overlapping patches $G_{j, j \in [1,2,...,m]}$ and $S_{j, j \in [1,2,...,m]}$, using a window of size $(ws, ws)$. Then $mask^i$ is get through:

$$mask_j^i = \begin{cases} 1, & mean(G_j) < \theta_l \& mean(S_j) < \theta_s \\ 0, & else \end{cases} \qquad (7)$$

where $mean$ denotes taking the average value, $mask_j^i$ denotes the $j$-th patch of $mask^i$, $\theta_l$ and $\theta_s$ are thresholds based on luminance distribution and SNR distribution respectively. We set $\theta_l$ and $\theta_s$ as the last 10% of luminance distribution and the last 30% of SNR distribution respectively. The extremely dark region feature of images can be defined as $F_{Id}^i = F_I^i \times mask^i$.

Difference from previous mask-based method [37], we integrate both luminance and SNR rather than solely focusing on one, and employ adaptive thresholds, which grant it superior generalization capability. Additionally, our mask operates at the patch level rather than the pixel level, which contributes more effectively to the fusion of images and events.

**Event-Image Attention** Most real datasets suffer from alignment issues between videos and events, we utilize a novel Event-Image Attention fusion to fuse them, and establish spatial mapping relationships to faciliate the alignment. Specifically, given the image feature $F_I^i \in R^{h \times w \times c}$, dark regions feature $F_{Id}^i \in R^{h \times w \times c}$ and event feature $F_E^i \in R^{h \times w \times c}$, we use $1 \times 1$ conv to get queries $Q_{id}$ from $F_{Id}^i$,

$Q_i$ from $F_I^i$, keys $K_e$ and values $V_e$ from $F_E^i$ respectively. Then we divide them into patches using a window of size $(ws, ws)$. The shape of features are changed from $(c, h, w)$ to $(h/ws \times w/ws, ws \times ws \times c)$, where each patch represents features from different spatial positions. The aggregated event feature $F_{Ef}^i$ is written as:

$$F_{Ef}^i = \Phi(Q_{id}, K_e, V_e) = V_e Softmax(\frac{Q_{id}^T K_e}{\sqrt{d_k}}), \qquad (8)$$

$d_k$ denotes the dimension of the features. It makes event feature focusing on extremely dark areas. To establish the spatial mapping relationship, we calculate an attention similarity between $F_I^i$ and $F_E^i$:

$$W_p = Q_i^T K_e \qquad (9)$$

which serves as a measure of the similarity between every image patch and event patch. For misaligned image and event features, we consider an event patch with the highest similarity to an image patch to be one that matches the image patch. Consequently, we set the maximum value of each row in $W_p$ to 1 while the remaining values to 0, and utilize it to establish a spatial mapping between two types of patches via:

$$\widetilde{F_{Ef}^i} = W_p^T F_{Ef}^i \qquad (10)$$

where $\widetilde{F_{Ef}^i}$ represents the event features aligned with the image features. Finally, the output of EIF is:

$$F_f^i = \widetilde{F_{Ef}^i} + F_I^i \qquad (11)$$

## 3.4 Temporal Consistency Loss

An event stream $E_{t-\Delta t \sim t+\Delta t}$ records dynamic information between $t-\Delta t$ and $t+\Delta t$. When $\Delta t \to 0$, it indicates a very small motion and reflects the motion trend. To better keep the temporal stability of output frames, we design the temporal consistency Loss $L_t$, which estimates the motion trend at time $t$ from frame $I_t$ and compares it with the input. For the synthetic data, the input events are aligned with the frames, and the temporal consistency loss is:

$$L_t^s = \| v(\widetilde{Y_t}) - E_{t-\Delta t \sim t+\Delta t} \|_F^2, \Delta t \to 0 \qquad (12)$$

For the real data, the input events and the frames are misaligned, the temporal consistency loss is:

$$L_t^r = \| v(I_t) - v(\widetilde{Y_t}) \|_F^2 \qquad (13)$$

where v denotes a U-Net [28] structure used to extract motion trend from frames.

## 3.5 Detail Contrast Loss

The detailed information in extremely dark areas is lost while relatively complete in other areas. To restore complete details in the enhanced frame, we encourage greater discrepancies between input frames and output frames in extremely dark regions, while reducing discrepancies in other areas. The loss function is defined as follows:

$$L_{con} = \frac{\| (I_t - \widetilde{Y_t})(1 - mask) \|_F^2}{\| (I_t - \widetilde{Y_t})(1 - mask) \|_F^2 + \| (I_t - \widetilde{Y_t})mask \|_F^2} \qquad (14)$$

where $mask$ identifies the extremely dark region.

Furthermore, we also employ the $L_{clip}$ in [17] to preserve the semantic integrity of the enhanced results.

$$L_{clip} = \frac{e^{cos(\Phi_{image}(\widetilde{Y}_t), \Phi_{text}(T_n))}}{\sum_{i \in \{n,p\}} e^{cos(\Phi_{image}(\widetilde{Y}_t), \Phi_{text}(T_i))}}$$
$$+ w \cdot \parallel \Phi_{image}(\widetilde{Y}_t) - \Phi_{image}^l(I_t) \parallel_2 \quad (15)$$

where $\Phi_{image}$ and $\Phi_{text}$ denote the image encoder and text encoder in CLIP model respectively, $w$ is the weight. We set $T_n$ = "normal light image" and $T_p$ = "low light image".

## 3.6 Optimization

We propose to train the REN over a partially labeled dataset, composed of the synthetic dataset $D_s$ with groundtruth and the real dataset $D_r$ without groundtruth. We employ $L_t^s$, $L_{error}$ for $D_s$, and $L_t^r$, $L_{con}$, $L_{clip}$ for $D_r$. $L_{error}$ is the reconstruction loss according to Eq. 5:

$$L_{error} = \parallel \widetilde{Y}_t - I_n \parallel_F^2 + \parallel \phi_E(I_n, E) - \phi_E(\widetilde{Y}_t, E) \parallel_F^2 \quad (16)$$

Thus the overall function is as follows:

$$L = L_{error} + \alpha L_t^s + \beta L_t^r + \gamma L_{con} + \delta L_{clip} \quad (17)$$

where $\alpha$, $\beta$, $\gamma$ and $\delta$ denote the balancing parameters.

# 4 EXPERIMENTS

## 4.1 Experimental Settings

**Dataset** The proposed Real-Event Embedded Network is trained in a semi-supervised strategy, where one synthetic dataset (SDSD dataset [34]) is provided for training with the groundtruth and one real dataset (DSEC dataset [6]) for unsupervised training. In addition, we use the ViViD++ dataset [14, 15] for cross-dataset evaluation.

*SDSD*: Based on the SDSD dataset [34], we build a syntheic dataset composed of synthetic events as well as real low-light videos and normal-light videos. We employ V2E [10] to simulate events from low-light videos. The detailed configuration of V2E can be found in the **supplementary material**.

*DSEC*: DSEC dataset [6] consists of real videos and events captured by Prophesee Gen3.1M camera. We select low-light videos along with events from it to train and test our method. We choose 10 training videos with a total of 2948 frames, along with 19 test videos totaling 570 frames.

*ViViD++*: Similar to the DSEC dataset, ViViD++ dataset consists of real videos and events captured by DVXplorer camera. We select low-light videos along with events from it for cross-dataset evaluation. Due to ViViD++ is collected from real-world scenarios, it reflects the effectiveness of the method in real-world settings.

**Implementation Details** Our network is implemented using Pytorch on NVIDIA Geforce RTX 3090. During training, we randomly crop the samples into $320 \times 320$ patches and augment the data using rotation and horizontal flipping. We optimize the network by AdamW optimizer [19] with the momentum terms of (0.9, 0.999). We set the learning rate to 4e-4 and use the cosine decay strategy to decrease it. The total epoch of training is set to 100 and the balancing parameters $\alpha$, $\beta$, $\gamma$, $\delta$ and $w$ are all set to 1. To validate the effectiveness of exploiting real event, REN is trained

respectively over two different datasets: SDSD and SDSD + DSEC, and final networks are respectively denoted as $REN_{sd}$ and REN. Specifically, $REN_{sd}$ is trained only over syntheic SDSD dataset, while REN is trained over syntheic SDSD dataset and real DSEC dataset.

## 4.2 Quantitative Evaluation

To comprehensively evaluate the effectiveness of our proposed method, we compare REN with SOTA methods using the SDSD dataset and DSEC dataset.

**Evaluation on SDSD dataset.** We train and test the SOTA methods on SDSD dataset for quantitative evaluations, the testset of SDSD dataset includes 12 indoor video pairs and 13 outdoor video pairs.

We adopt three evaluation metrics: Peak Singal-to-Noise Ratio (PSNR), Structural Similarity (SSIM) [35] and Learned Perceptual Image Patch Similarity (LPIPS) [42]. PSNR quantifies image quality by comparing signal power to noise. SSIM assesses the image similarity by considering luminance, contrast and structure. LPIPS measures the perceptual image patch similarity using deep learning.

The results are shown in Table 1, the $*$ in table means unsupervised methods. As these methods can be trained on SDSD dataset or SDSD+DSEC datasets, which is similar to ours, we only present the best results of these methods in the table, and the full results are presented in the **supplementary materials**. As shown in the table, $REN_{sd}$ outperforms all supervised methods and REN achieves the best performance. Furthermore, the application of real events and our semi-supervised training strategy is shown to be very effective in improving the performance of the model.

**Evaluation on DSEC dataset.** We also compare different methods on the DSEC real dataset. For supervised methods, due to the lack of groundtruth in the DSEC dataset, it is not feasible to them. Therefore, to compare them with our method, we generate results using models trained on the SDSD dataset, which is consistent with the $REN_{sd}$. For the unsupervised methods (marked with $*$), we train them on SDSD+DSEC datasets and compare them with REN.

We employ four no-reference evaluation metrics: Natural Image Quality Evaluator (NIQE) [24], Integrated Local NIQE (ILNIQE) [41], Perceptual Index (PI) [22] and Perception based Image Quality Evaluator (PIQE) [32]. NIQE assesses the image naturalness based on statistical properties. ILNIQE is an opinion-unaware blind image quality assessment method. PI evaluates the image quality based on human perceptual assessment. PIQE is based on the perceptual characteristics of the human visual system. The evaluation results are shown in Table 2, under the same training dataset conditions, $REN_{sd}$ outperforms all supervised methods, while REN outperforms all unsupervised methods among all the metrics.

We also integrate event to two recently best low-light video enhancement methods, SDSDNet [34] and LAN [4], for a more fair comparison. They are represented as SDSDNet + Event and LAN + Event respectively. Specifically, we incorporate the EIF module into them and train them on SDSD dataset. As shown in Table 3, both methods improve the performance after integrated with events, and our method achieves the best results.

## 4.3 Qualitative Evaluation

We perform qualitative assessments on the SDSD and DSEC datasets, and cross-dataset evaluations on ViViD++ dataset. Fig. 4 presents

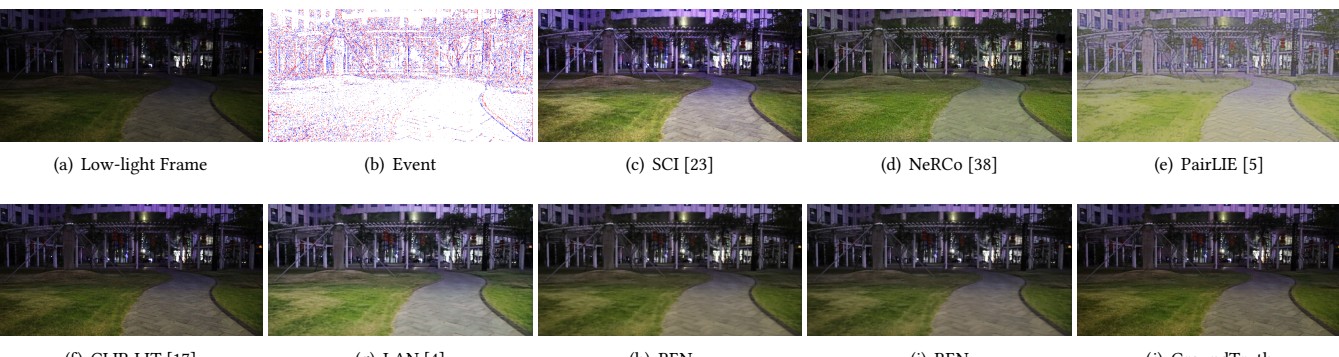

**Figure 4: Visual quality comparison of enhancement results on SDSD synthetic dataset.**

**Table 1: Quantitative results of different methods on SDSD dataset, ∗ means unsupervised methods.**

| Methods | Learning | PSNR↑ | SSIM↑ | LPIPS↓ |
|---|---|---|---|---|
| PairLIE [5] | Image | 13.37 | 0.54 | 0.22 |
| SCI* [7] | Image | 19.67 | 0.69 | 0.29 |
| SNR-Aware [37] | Image | 25.28 | 0.82 | 0.13 |
| NeRCo* [38] | Image | 20.07 | 0.66 | 0.21 |
| CLIP-LIT* [17] | Image | 22.29 | 0.72 | 0.16 |
| MBLLEN [21] | Video | 21.79 | 0.65 | 0.19 |
| SGLLIE* [43] | Video | 23.89 | 0.70 | 0.31 |
| SMID [1] | Video | 24.09 | 0.69 | 0.21 |
| SDSDNet [34] | Video | 24.92 | 0.73 | 0.14 |
| LAN [4] | Video | 27.25 | 0.85 | 0.12 |
| $REN_{sd}$ | Video | 28.45 | **0.88** | **0.09** |
| REN | Video | **29.03** | **0.88** | **0.09** |

**Table 2: Quantitative results of different unsupervised methods on DSEC dataset, ∗ means unsupervised methods.**

| Methods | NIQE↓ | ILNIQE↓ | PI↓ | PIQE↓ |
|---|---|---|---|---|
| PairLIE [5] | 12.08 | 33.66 | 3.80 | 20.59 |
| SNR-Aware [37] | 11.58 | 23.53 | 3.52 | 8.16 |
| MBLLEN [21] | 13.07 | 23.07 | 3.32 | 19.18 |
| SMID [1] | 20.89 | 26.88 | 9.10 | 13.22 |
| SDSDNet [34] | 14.16 | 34.93 | 5.64 | 15.47 |
| LAN [4] | 22.30 | 33.66 | 3.80 | 23.59 |
| $REN_{sd}$ | 10.84 | 20.76 | 3.26 | 15.65 |
| SCI* [23] | 11.70 | 21.30 | 3.41 | 8.35 |
| NeRCo* [38] | 13.64 | 22.63 | 2.71 | 14.70 |
| CLIP-LIT* [17] | 10.42 | 22.89 | 3.43 | 9.64 |
| SGLLIE* [43] | 11.07 | 24.11 | 3.77 | 8.57 |
| REN | **10.28** | **20.69** | **2.69** | **8.01** |

the results obtained on the SDSD dataset, similar to Table 1, we present the best results of unsupervised methods. In comparison to the groundtruth, the CLIP-LIT produces results with weak enhancement and low brightness. The PairLIE generates unrealistic outcomes. NeRCo, SCI and LAN suffer from significant color deviation, which adversely impairs the visual quality of the images.

Fig. 5 presents the visual results obtained on the DSEC dataset, which comprises low-quality videos with numerous extremely dark

**Table 3: Quantitative results of two recently best low-light video enhancement methods integrated with event on SDSD dataset.**

| Methods | PSNR↑ | SSIM↑ | LPIPS↓ |
|---|---|---|---|
| SDSDNet [34] | 24.92 | 0.73 | 0.14 |
| SDSDNet + Event | 25.81 | 0.85 | 0.11 |
| LAN [4] | 27.25 | 0.85 | 0.12 |
| LAN + Event | 28.20 | 0.87 | 0.10 |
| $REN_{sd}$ | **28.45** | **0.88** | **0.09** |

regions, posing a considerable challenge for enhancement. All the method are trained on SDSD+DSEC datasets. The SCI and CLIP-LIT result in significantly reduced brightness and severe noise, they are unable to effectively recover details in extremely dark areas. The result generated by NeRCo exhibits noticeable striping artifacts. The SGLLIE exhibits severe white noise and generates unrealistic results. Due to training exclusively on synthetic dataset, $REN_{sd}$ exhibits issues of insufficient brightness and unclear details on real data. In contrast, REN is trained on real data and demonstrates favorable results in enhancing details, improving brightness, and mitigating noise in extremely dark regions.

Fig. 6 shows the visual results obtained on the ViViD++ dataset. These results are generated using the models presented in Table 2. The PairLIE enhances the brightness but generates unrealistic color tones. The images enhanced by NeRCo and SDSDNet exhibit noticeable artifacts. CLIP-LIT and SCI fail to enhance the brightness adequately, while the result enhanced by LAN appears relatively blurry. The image enhanced by $REN_{sd}$ lacks sufficient clarity and exhibits blurriness in the details. In contrast, REN generates more realistic and clearer result.

Fig. 7 shows the visual results of SDSDNet and LAN as well as their integration with event, it also shows the result of our method. Upon the integration with event, both SDSDNet and LAN exhibit noticeable improvements in detail recovery and visual effects, demonstrating the efficacy of event and EIF module. Meanwhile, $REN_{sd}$ and REN achieve the most favorable visual results.

After a comprehensive evaluation of the comparative results of various methods across the three datasets, our proposed approach demonstrates outstanding visual performance in global brightness and detail restoration. More experimental results can be found in the **supplementary material**.

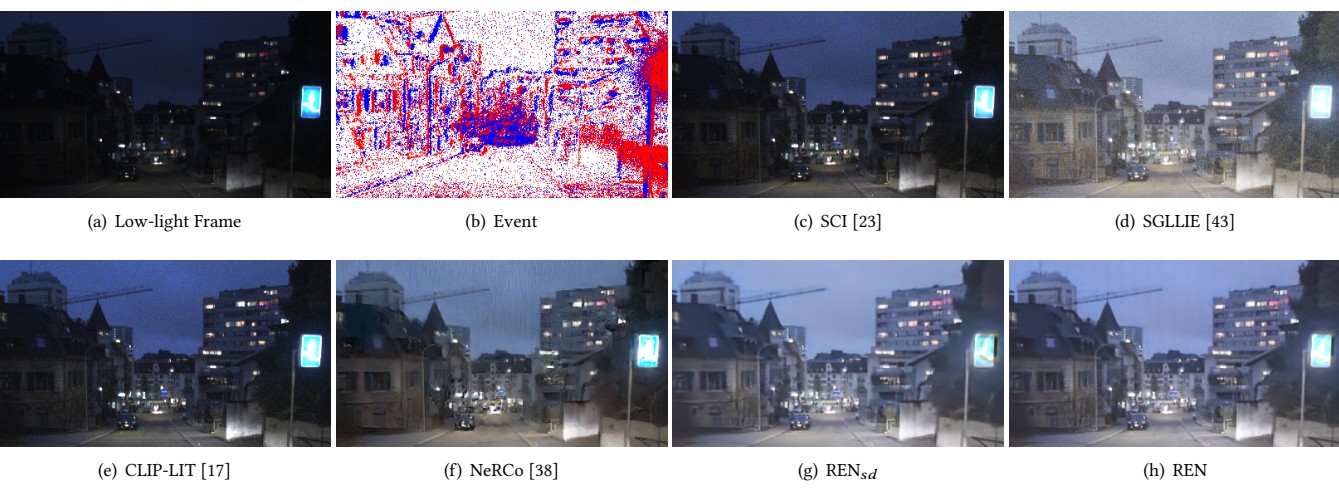

(a) Low-light Frame  (b) Event  (c) SCI [23]  (d) SGLLIE [43]

(e) CLIP-LIT [17]  (f) NeRCo [38]  (g) REN$_{sd}$  (h) REN

**Figure 5: Visual quality comparison of enhancement results on DSEC real dataset.**

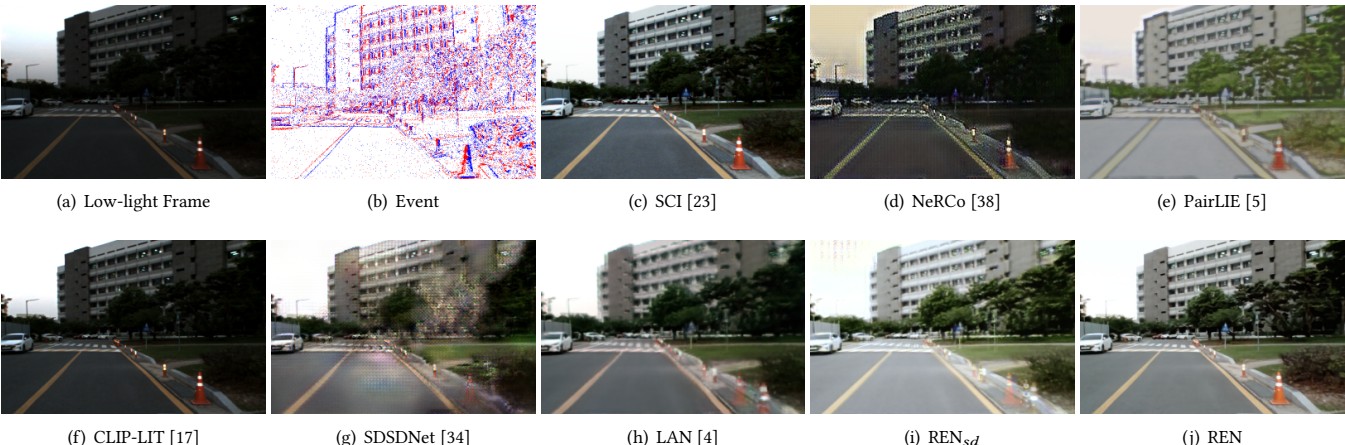

(a) Low-light Frame  (b) Event  (c) SCI [23]  (d) NeRCo [38]  (e) PairLIE [5]

(f) CLIP-LIT [17]  (g) SDSDNet [34]  (h) LAN [4]  (i) REN$_{sd}$  (j) REN

**Figure 6: Visual quality comparison of enhancement results on ViViD++ real dataset.**

## 4.4 Ablation Study

The proposed REN improves the performance of low-light video enhancement. To find out what contributes to the remarkable effectiveness of our approach, we do an ablation study and show the results in Table 4 and Fig. 8.

**The effectiveness of Events.** To validate the effectiveness of events, we train the model only using videos (Only Video). The PSNR value of this model decreases sharply by 2.91dB, which indicates the significant impact of events. As shown in Fig. 8, events significantly contribute to the recovery of details. The effectiveness of real events (REN$_{sd}$, W/o Real Events) is already demonstrated in Table 1 and Table 2, where REN outperforms REN$_{sd}$ on all datasets, and as shown in Fig. 8, REN achieves superio results on real data compared to REN$_{sd}$ (W/o Real Events).

**The effectiveness of EIF Module.** The EIF module consists of two parts, and concurrently establishes the mapping relationship between images and events, so we build three modified models for ablation study. 1) W/o Dark Region Mask (W/o DRM), which means we remove Dark Region Mask from the model. 2) W/o Event

**Table 4: Quantitative results of ablation study on SDSD test-set.**

| Model | PSNR↑ | SSIM↑ | LPIPS↓ |
|---|---|---|---|
| Only Video | 26.12 | 0.86 | 0.10 |
| REN$_{sd}$ (W/o Real Events) | 28.45 | 0.88 | 0.09 |
| W/o DRM | 27.15 | 0.87 | 0.11 |
| W/o EIA | 26.29 | 0.87 | 0.09 |
| W/o $W_p$ | 27.63 | 0.87 | 0.09 |
| W/o $L_t$ and $L_{con}$ | 28.61 | 0.88 | 0.09 |
| W/o $L_t$ | 28.80 | 0.88 | 0.09 |
| W/o $L_{con}$ | 28.91 | 0.88 | 0.09 |
| REN | **29.03** | 0.88 | 0.09 |

Image Attention (W/o EIA), which means we replace the Event Image Attention with a U-Net, as Dark Region Mask is designed to complement Event Image Attention, we also remove it. 3) W/o $W_p$, which means we remove $W_p$ and Eq. 10 from EIF Module, rendering it incapable of establishing spatial alignment between images and

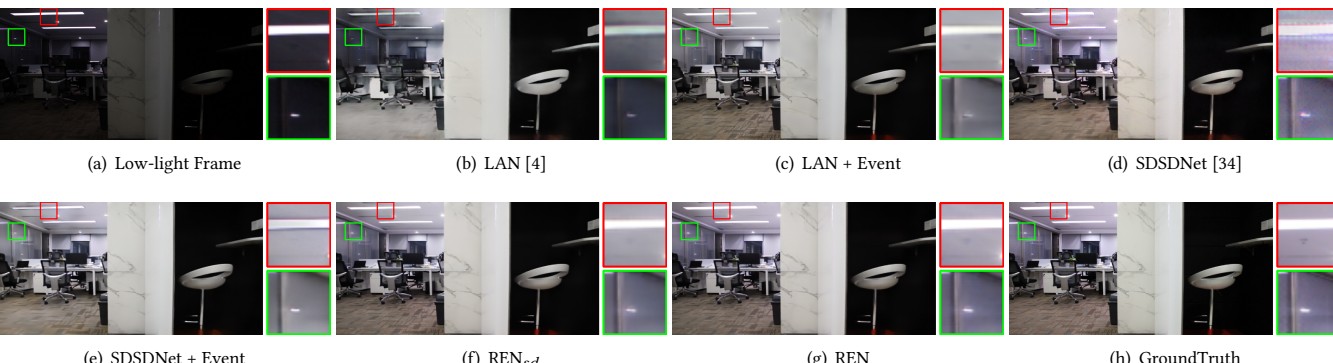

(a) Low-light Frame      (b) LAN [4]      (c) LAN + Event      (d) SDSDNet [34]

(e) SDSDNet + Event      (f) $REN_{sd}$      (g) REN      (h) GroundTruth

**Figure 7: Visual results of SDSDNet [34] and LAN [4] as well as their integration with event.**

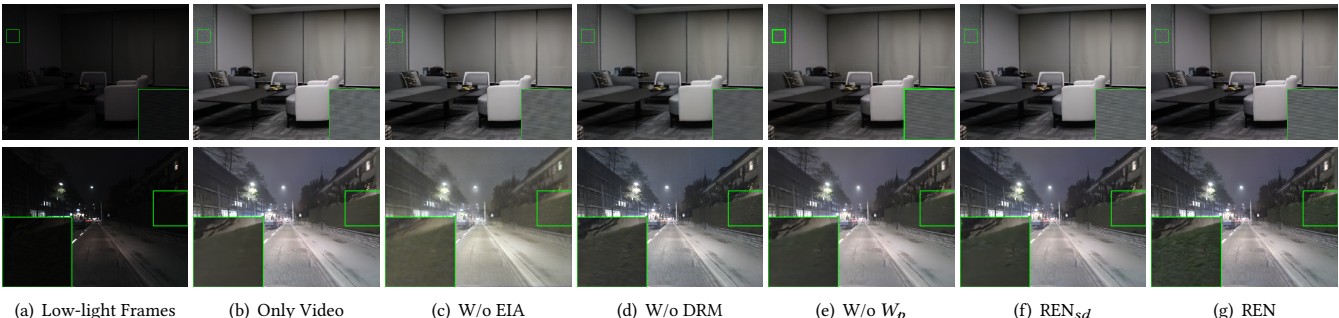

(a) Low-light Frames    (b) Only Video    (c) W/o EIA    (d) W/o DRM    (e) W/o $W_p$    (f) $REN_{sd}$    (g) REN

**Figure 8: Visual result of our ablation study. The top row is synthetic data from SDSD dataset and the bottom row is real data from DSEC dataset.**

**Table 5: The results of the user study. "REN" means that our result is preferred, "Other" means some other approach is preferred, "Same" means that the users have no preference.**

| Methods | Other | Same | REN |
|---|---|---|---|
| MBLLEN [21] | 15% | 5% | 80% |
| SGLLIE [43] | 5% | 0% | 95% |
| SMID [1] | 0% | 0% | 100% |
| SDSDNet [34] | 5% | 5% | 90% |
| LAN [4] | 15% | 25% | 60% |

events. The PSNR value of these models drop 1.88dB, 2.74dB and 1.4dB respectively. As shown in Fig. 8, the models W/o DRM and W/o EIA demonstrate unsatisfactory performance in the recovery of details within extremely dark regions. Due to the inability to align with real events and images, the model W/o $W_p$ fails to fully leverage event information and recover details on real data. The result shows the effectiveness of our EIF module. More detailed discussions regarding the network architecture can be found in the **supplementary material**.

**The effectiveness of temporal consistency loss and detail contrast loss.** To validate the effectiveness of the proposed $L_t$ and $L_{con}$, we build three modified strategies: 1) W/o $L_t$ and $L_{con}$, which means training the model only using $L_{clip}$ and $L_{error}$. 2) W/o $L_t$, which means training the model without $L_t$. 3) W/o $L_{con}$, which means training the model without $L_{con}$. The PSNR value of these models drop 0.42dB, 0.23dB, 0.12dB respectively, which shows the effectiveness of our proposed $L_t$ and $L_{con}$.

## 4.5 User Study

We conduct user studies with 20 participants to compare the subjective visual quality of REN and other low-light video enhancement methods, including MBLLEN [21], SGLLIE [43], SMID [1], SDSD-Net [34], LAN [4]. We randomly select 10 enhanced videos from the enhancement results of ViViD++ [14, 15] dataset to compare the performance of different models.

Each participant underwent five sets of test. In each set, participants are required to randomly select one method from the five options excluding REN and then choose a video randomly. They are instructed to compare the enhancement results of the selected method with REN on that particular video, and then they should decide which is better. The quantitative results of the user studies are shown in Table 5, indicating that our method is more appealing to users compared to other approaches.

## 5 CONCLUSION

In this work, to address the issue of information loss in extremely dark regions on low-light video enhancement, we propose the Real-Event Embedded Network, which leverages events to enhance such regions in low-light videos. We also design unsupervised temporal consistency loss and detail contrast loss, aiming to maintain temporal stability of videos and restore details in extremely dark regions respectively. These two loss functions, along with the supervised loss, are jointly employed in a semi-supervised manner to train the network on real data. Extensive experiments demonstrate that our method achieves SOTA performance.

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
