# OpenReview forum: "Exploring in Extremely Dark: Low-Light Video Enhancement with Real Events"
_acmmm.org/ACMMM/2024/Conference — MM2024 Poster_

### Official Review · Reviewer_7vyw · 2024-05-24

**Rating:** 5
**Confidence:** 1

**Summary:**

The paper presents a novel approach to enhance low-light video using event cameras, which provide high dynamic range and can capture motion information even in darkness. The authors propose the Real-Event Embedded Network (REN), which integrates real event data to enhance low-light videos. The approach includes an Event-Image Fusion (EIF) module that identifies and enhances dark regions using event information. The REN employs both supervised and unsupervised loss functions to ensure temporal stability and detail restoration, demonstrating superior performance on both synthetic and real datasets compared to existing methods.

**Strengths:**

1.  The use of real events to enhance low-light video is innovative and effective based on the authors' presentation.

2. The design of unsupervised temporal consistency loss and detail contrast loss ensures the temporal and spatial quality of the enhanced videos.

3. The paper includes thorough experimental validation on both synthetic and real datasets, demonstrating the superiority of their method over state-of-the-art approaches and the applicability of the proposed method in the real-world scenario.

**Limitations:**

1. I notice the real-event, low-light and ground-truth data used in this paper is paired, which may cause challenges in obtaining more data and the broader applicability and generalization of this method may be undermined.

2. I am not familiar with real-event cameras, but could the proposed method, though trained with real-events, be used in the data captured by traditional cameras? I look forward to the authors' reply to clear my confusions.

3. To better demonstrate the applicability of the proposed method, some downstream vision tasks, like detection and segmentation, can be conducted on the enhanced results.

**Suitability:**

2

---

### Official Review · Reviewer_Xb1v · 2024-05-25

**Rating:** 1
**Confidence:** 3

**Summary:**

- This paper proposes an Event-Image Fusion module, which can identify these dark regions and enhance them significantly.
- To ensure temporal stability of the video and restore details within extremely dark areas, this paper design unsupervised temporal consistency loss and detail contrast loss.

**Strengths:**

- To ensure temporal stability of the video and restore details within extremely dark areas, this paper design unsupervised temporal consistency loss and detail contrast loss.

**Limitations:**

1. Figure 2 has the problem that the data streamlines are too thin and the fonts are messy, which is not easy for readers to read and understand. Please ask the author to optimize the fonts and data streamlines.
2. Basis for the selection of loss function hyperparameters? Did the experiment compare? With all parameters set to 1, what is the point of introducing parameters? In addition, what is the basis for hyperparameter selection?
3. What is the meaning of the second line of formula 5? In this paper, the reflection estimation obtained from image combined with event camera is restricted to only using image to get reflection estimation. What is the specific meaning of adding event camera? Why add it?
4. In Method 3.4, how to use U-Net to extract motion trends in line 450? The real data time is not aligned, how to extract the movement trend, how to maintain the time consistency? Please add the specific formula and explain the specific principle.
5. The contrast method lacks the latest low-light image enhancement method; As shown in the experimental results in Figure 4, the enhanced floor tile details of the proposed method have fuzzy problems, which are obviously inferior to the SCI algorithm. Similarly, the tower crane in Figure 5 has the same problem. We think it is caused by less information on the edge of the event camera. How should the author solve and improve the above problems?
6. Please add a brief introduction to the comparison method and explain why the above method was chosen.
7. In Figure 2, why do the two modules proposed in this paper need to be repeated four times? What is the basis of the experiment? Please supplement the experimental description.

**Suitability:**

2

---

### Official Review · Reviewer_VHaJ · 2024-06-05

**Rating:** 4
**Confidence:** 3

**Summary:**

This paper presents an event-based low-light video enhancement method. An event-image fusion module is proposed to primarily utilize the given events to enhance the dark regions of RGB images. More deeply, novel event-image attention is designed to establish a relationship between the event features and dark-region image features. Furthermore, since there is no real-world normal-light gt of low-light videos with real-world events, the paper proposes to train the method with a supervised strategy on the synthetic event data and an unsupervised strategy on the real event data. An unsupervised detail contrast loss is proposed to primarily enhance the dark regions of input images. The proposed method achieves state-of-the-art performance on the synthetic test set and the real-world test set.

**Strengths:**

+ This paper presents a novel event-image attention for enhancing dark-region features. Additionally, an unsupervised detail contrast loss is proposed to improve the dark-region low-light enhancement;
+ This paper is well-written, well-organized and easy to comprehend.
+ The proposed method achieves state-of-the-art performance both on quantiative and qualitative evaluations.

**Limitations:**

- Missing a necessary baseline. Although authors mention the paper ([ICCV2023] Coherent Event Guided Low-Light Video Enhancement) in the "Related Work", they do not compare the proposed method with this competitive baseline;

**Suitability:**

3

---

### Official Review · Reviewer_Govf · 2024-06-09

**Rating:** 5
**Confidence:** 3

**Summary:**

The paper proposes a novel network to use event signals guide low-light video enhancement.
Details contained in event signals are extracted by segmenting extreme dark areas and integrated into contrast enhanced RGB images.
Multiple loss functions are adopted to facilitate more robust results.
Comprehensive experiments are carried out to prove advantages brought about by the proposed method.

**Strengths:**

- Thorough theoretical analysis on image formation and pipeline design. Both dark region mask estimation and event-image attention module have alignments with intuition as well as detailed reasoning to support design choices.
- Comprehensive experiment results presented. Including both synthetic and real-world datasets, detailed ablation studies, and user study. All showing convincing performance improvements compared to existing methods.

**Limitations:**

There is one minor concern regarding the generalization ability of the proposed method. With a large number of parameters and obvious performance gain, it is natural to ask about how well the proposed model can generalize to unseen data. A cross-dataset test experiment could be helpful.

**Suitability:**

2

---

### Meta-Review · Area_Chair_3p6g · 2024-07-02

**Recommendation:** Accept (Poster)
**Confidence:** 4

**Metareview:**

Most reviewers find the presented method has technical strengths, with the proposed event-image fusion attention module. The experimental results demonstrate convincing improvement over prior work, with comprehensive results on different datasets and ablation studies. The authors are expected to supplement more experimental results to further address some questions. The AC recommends acceptance to this paper.